# Exposure to mold proteases stimulates mucin production in airway epithelial cells through Ras/Raf1/ERK signal pathway

**Xianxian Wu**[1,2☯]**, Boram Lee**[2☯]**, Lingxiang Zhu**[2]**, Zhi Ding**[1,3]***, Yin Chen**[2,4]***

**1** State Key Laboratory of Pharmaceutical Biotechnology, School of Life Sciences, Nanjing University, Nanjing, Jiangsu, China, **2** Department of Pharmacology and Toxicology, School of Pharmacy, University of Arizona, Tucson, Arizona, United States of America, **3** Changzhou High-Tech Research Institute of Nanjing University, Changzhou, China, **4** Asthma & Airway Disease Research Center, University of Arizona, Tucson, Arizona, United States of America

☯ These authors contributed equally to this work.
* dingzhi@nju.edu.cn (ZD); ychen@pharmacy.arizona.edu (YC)

**Data Availability Statement:** All relevant data are within the manuscript and its Supporting Information files.

## Abstract

Environmental mold (fungus) exposure poses a significant threat to public health by causing illnesses ranging from invasive fungal diseases in immune compromised individuals to allergic hypertensive diseases such as asthma and asthma exacerbation in otherwise healthy people. However, the molecular pathogenesis has not been completely understood, and treatment options are limited. Due to its thermo-tolerance to the normal human body temperature, *Aspergillus. fumigatus* (*A.fumigatus*) is one of the most important human pathogens to cause different lung fungal diseases including fungal asthma. Airway obstruction and hyperresponsiveness caused by mucus overproduction are the hallmarks of many *A.fumigatus* induced lung diseases. To understand the underlying molecular mechanism, we have utilized a well-established *A.fumigatus* extracts (AFE) model to elucidate downstream signal pathways that mediate *A.fumigatus* induced mucin production in airway epithelial cells. AFE was found to stimulate time- and dose-dependent increase of major airway mucin gene expression (MUC5AC and MUC5B) partly *via* the elevation of their promoter activities. We also demonstrated that EGFR was required but not sufficient for AFE-induced mucin expression, filling the paradoxical gap from a previous study using the same model. Furthermore, we showed that fungal proteases in AFE were responsible for mucin induction by activating a Ras/Raf1/ERK signaling pathway. $Ca^{2+}$ signaling, but ROS, both of which were stimulated by fungal proteases, was an indispensable determinant for ERK activation and mucin induction. The discovery of this novel pathway likely contributes to our understanding of the pathogenesis of fungal sensitization in allergic diseases such as fungal asthma.

## Introduction

*Aspergillus* is a group of molds with around 200 species commonly found both indoor and outdoor [1, 2]. *Aspergillus* is one of the most common indoor molds according to National

**Funding:** The study was supported by Arizona Biomedical Research Commission innovative award BIG- 3064 and the funding from College of Pharmacy at University of Arizona. The funders had no role in study design, data collection and analysis, decision to publish, or preparation of the manuscript. There was no additional external funding received for this study.

**Competing interests:** The authors have declared that no competing interests exist.

Institute of Environment Health Science (NIEHS) [3] and Centers for Disease Control and Prevention (CDC) [4] They grow in damp soils, decaying vegetation, organic debris, and exist in bedding in houses [1, 2]. They are present in the atmosphere throughout the year, but the concentration peaks in late autumn [1]. Among these *Aspergillus* species, *Aspergillus fumigatus* (*A.fumigatus*) is one of the most common species that cause lung diseases, likely due to its unique tolerance to body temperatures [5]. One of the major illnesses caused by *A.fumigatus* is fungal asthma. The prelude of asthma development is usually a repeated environmental allergen (e.g. mold) exposure and sensitization leading to type 2 immune response (or T2IR) [6, 7]. Exposure to indoor molds including *Aspergillus* during the first 2 years of life was found to associate with an increased risk of developing asthma by the meta-analysis of 8 birth cohorts in Europe [8]. The prevalence of fungal sensitization in general asthmatics is high (28% on average and as high as 48%) [9]. Fungal asthma is oftentimes poorly managed with frequent exacerbations and hospitalizations [10–13]. Beside fungal asthma, *A.fumigatus* can also cause other severe fungal diseases such as aspergilloma, allergic bronchopulmonary aspergillosis (ABPA) and invasive aspergillosis [1, 14, 15] in individuals with a compromised immune system (e.g. AIDS, patients receiving transplant, or under immune-suppressive medications) [15]. In these individuals, *A.fumigatus* could spread from the initial site of infection in the lung to other organs and lead to fatal results [15, 16].

Interestingly, mucus overproduction is associated with almost all of *A.fumigatus* induced airway diseases including ABPA and fungal asthma. Airway obstruction caused by mucus overproduction and damage to the tracheobronchial walls are the hallmarks of bronchiectasis caused by *A.fumigatus* infection [17]. In asthma, mucus occlusion of small airway, and causes airway hyperresponsiveness, one of the major pathogenic factors [18]. Additionally, *A.fumigatus* exposure exacerbates existing chronic lung diseases including asthma, COPD or cystic fibrosis [19], in those diseases, mucus overproduction is a pathogenic hallmark leading to decreased lung function. The major macromolecular components of mucus are high-molecular-weight polymeric gel-forming mucin glycoproteins. In airway, the major gel-forming mucins are MUC5AC and MUC5B [20, 21]. The mechanism underlying *A.fumigatus* induced mucin production has not been well studied. *A. fumigatus* extracts (AFE) was previously reported to induce MUC5AC mRNA and protein expression in airway epithelial cells through the activation of epidermal growth factor receptor (EGFR) [22]. However, in that study, although EGFR inhibitors could effectively block AFE induced MUC5AC expression, a direct EGFR activation by AFE was not demonstrated [22]. In our present study using the exact same epithelial cell culture model, we made a surprising discovery that AFE did not enhance EGFR activity, despite the fact that this activity was required for mucin induction. Instead, AFE increased Ras/Raf1/ERK pathway that was likely responsible for mucin induction in the epithelial cells.

## Materials and methods

### Materials

*A. fumigatus* extracts (AFE) was purchased from GREER (Lenoir, NC). AG1478 (Sigma, St. Louis, MO), BIBX 1382 (Sigma, St. Louis, MO), neutralizing anti-EGFR antibody (Calbiochem, La Jolla, CA), Raf-1 inhibitor (Sigma, St. Louis, MO) and sorafenib (LC laboratories, Woburn, MA), U0126 (1,4-diamino-2,3-dicyano-1,4-bis [2-aminophenylthio] butadiene) (Sigma, St. Louis, MO). PMSF and Glutathione reduced ethyl ester (GSH-MEE) were from Sigma (St. Louis, MO) and phosphatase inhibitor was from Thermo Fisher Scientific (Waltham, MA). Antibodies targeting pERK1/2, pRaf-1, and anti-PAR2 neutralizing antibody were purchased from Cell Signaling Technology (Danvers, MA). Anti-MUC5B, anti-pEGFR

(Y1173), anti-EGFR, anti-pTyr and β- ACTIN antibodies were from Santa Cruz Biotechnology (Santa Cruz, CA), Anti-MUC5AC and anti-Ras antibodies were obtained from Thermo Fisher Scientific (Grand Island, NY). Protease assay kit was obtained from Invitrogen (Carlsbad, CA).

## Cell culture

A human lung mucoepidermoid pulmonary carcinoma cell line, NCI-H292 [23], was obtained from ATCC (American Type Culture Collection ® CRL-1848™) (Manassas, VA). There was no reported misidentification or contamination based on the ICLAC Database. The authenticity of this cell line was further confirmed by a morphological assessment and the measurement of cell markers. All our cell cultures were routinely screened for mycoplasma contamination using a mycoplasma detection kit (Lonza, NJ). The cells were grown in RPMI 1640 medium supplemented with 10% FBS as described previously [24]. The cells were serum starved for 24 hours (hrs) before any treatment. All treatments were performed under a serum-free condition. For stimulation, 7.5 μg/ml AFE or 50 ng/ml EGF or 100 nM Trypsin were used to treat cells for different duration as indicated in the text. For protease inhibitor studies, protease inhibitors were mixed with AFE and pre-incubated for 15 minutes (min) in a cell culture incubator to remove the protease activity in the AFE. Each of inhibitor/AFE mixtures was added to cell cultures for 6 hrs. For inhibitory studies, cells were pre-treated with any inhibitor for 1 hr before the treatment.

## RNA isolation and qRT-PCR

RNA isolation procedures followed the TRIzol® Reagent manual (Invitrogen, Carlsbad, CA). Real-time PCR was performed as described previously [25]. The relative mRNA amount in each sample was calculated based on the $C_t$ method using housekeeping gene *Actin*. Results were usually calculated as fold induction over control [25]. All primers are shown in Table 1.

## Immunoprecipitation for phosphorylated EGFR

Cells were washed with PBS and lysed on ice with lysis buffer containing 0.5% Triton X-100, 70 mM β-glycerol phosphate, 1 mM EGTA, 1 mM DTT, 2 mM $MgCl_2$, 1:100 of protease inhibitor and 1:100 of phosphatase inhibitor. 2 μg of EGFR antibody per 100 μg cell lysate was added and incubated at 4˚C for overnight. 20 μl of Protein A-agarose beads (Sigma, St. Louis, MO), which bind to rabbit IgG, was then added and incubated for 1 hr at 4˚C. The proteins were immunoblotted with an anti-phosphoserine mAb (PY99, Santa Cruz Biotechnology, Santa Cruz, CA).

## Measurement of promoter activity by a luciferase assay

NCI-H292 cells were transfected with pGL3-MUC5AC [26] or pGL3-MUC5B [27] promoter reporter plasmid construct using Lipofectamine 2000 (Thermo Fisher Scientific, Grand Island,

**Table 1. Real-time primers.**

| Gene | Primer |
| --- | --- |
| qMUC5AC | Forward GCCTTCACTGTACTGGCTGAG |
| qMUC5AC | Reverse TGGGTGTAGATCTGGTTCAGG |
| qMUC5B | Forward CCCACTTCTCTACTCCCTGCT |
| qMUC5B | Reverse CTGATTGCACACTGCGTAGAA |
| qACTIN_ | Forward ACTGGAACGGTGAAGGTGACA |
| qACTIN_ | Reverse ATGGCAAGGGACTTCCTGTAAC |

NY). The transfected cells were treated with AFE at different doses as indicated for 24 hrs. Luciferase activity, the surrogate of promoter reporter, was performed by a Dual-Luciferase reporter assay from Promega (Madison, WI) and expressed as a fold induction above the control (luciferase reading from PGL3 empty vector transfected cells).

### Active Ras pull-down assay

Active Ras GTPase was detected and quantified using an active Ras pull-down and detection kit from Thermo Fisher Scientific (Grand Island, NY) according to the manufacturer's instruction. Briefly, GST-Raf1-RBD (Ras-binding domain), coupled to glutathione agarose resin, was used to pull down the activated Ras. Addition of GTP or GDP was used as a positive control or negative control. Eluted samples were then separated on a 12% acrylamide gel and analyzed using ani-Ras antibody provided in the kit.

### Measurement of intracellular calcium

Cells were grown in 35 mm glass-bottom dishes and were pre-treated with 50 μM of BAPTA (Sigma, St. Louis, MO) or PBS control for 30 min. Cells were then washed with OPTI-MEM followed by incubating with 5 μM Fluo-4/AM (Invitrogen, Carlsbad, CA) in the dark for 30 min in OPTI-MEM. After washing, the cellular fluorescence was imaged by a time lapse assay using confocal microscopy (LSM 510 meta; Carl Zeiss, Thornwood, NY).

### Statistical analysis

Graphpad Prism 8 [28] was used for all data analysis. The data was shown as Mean ± SEM. Experimental groups were compared using a two-sided Student's t test, with significance level set as $P < 0.05$. When data were not distributed normally, significance was assessed with the Wilcoxon matched-pairs signed-ranks test, and $P < 0.05$ was considered to be significant. The number of biological replicates for each experiment is described under Results and also in the figure legends.

## Results

### Heat-labile components of *Aspergillus fumigatus* extracts (AFE) significantly increased MUC5AC and 5B expression

MUC5AC and MUC5B transcripts were measured in epithelial cells reacted with three different concentrations of AFE for 6 and 24 hrs, respectively. A significant induction (~80-fold) of MUC5AC was observed even at the lowest concentration of AFE (1μg/ml) for a 24hr treatment. The elevation of MUC5AC was dose dependent (1–7.5 μg/ml) and reached more than 100-fold at 7.5 μg/ml (Fig 1A). MUC5AC mRNA increase was also time dependent. It occurred as early as 6 hrs after the treatment and continued its rise until 24 hrs later (Fig 1A). MUC5B, another major airway gel-forming mucin, was also increased by AFE treatment in the similar time- and dose-dependent manner to MUC5AC albeit by a less magnitude (Fig 1B). AFE treatment increased MUC5B mRNA by ~1.5–3.5 folds at 6 hrs and 4–10.5 folds at 24 hrs. Interestingly, the mucin-inducing activity of AFE was lost when heated in 72˚C for 30 min (Fig 1C), suggesting that heat-labile substance in AFE was responsible for mucin induction. Because of the robust induction of mucins at 6 hrs, we decided to focus our following mechanistic study at this time point to capture the early signaling event.

Because the increase of steady-state mRNA level may indicate the transcriptional regulation, we tested the activity of MUC5AC- and MUC5B-promoter luciferase reporter. AFE significantly increased the promoter activity of MUC5AC by ~1.5 folds and MUC5B by ~2.5

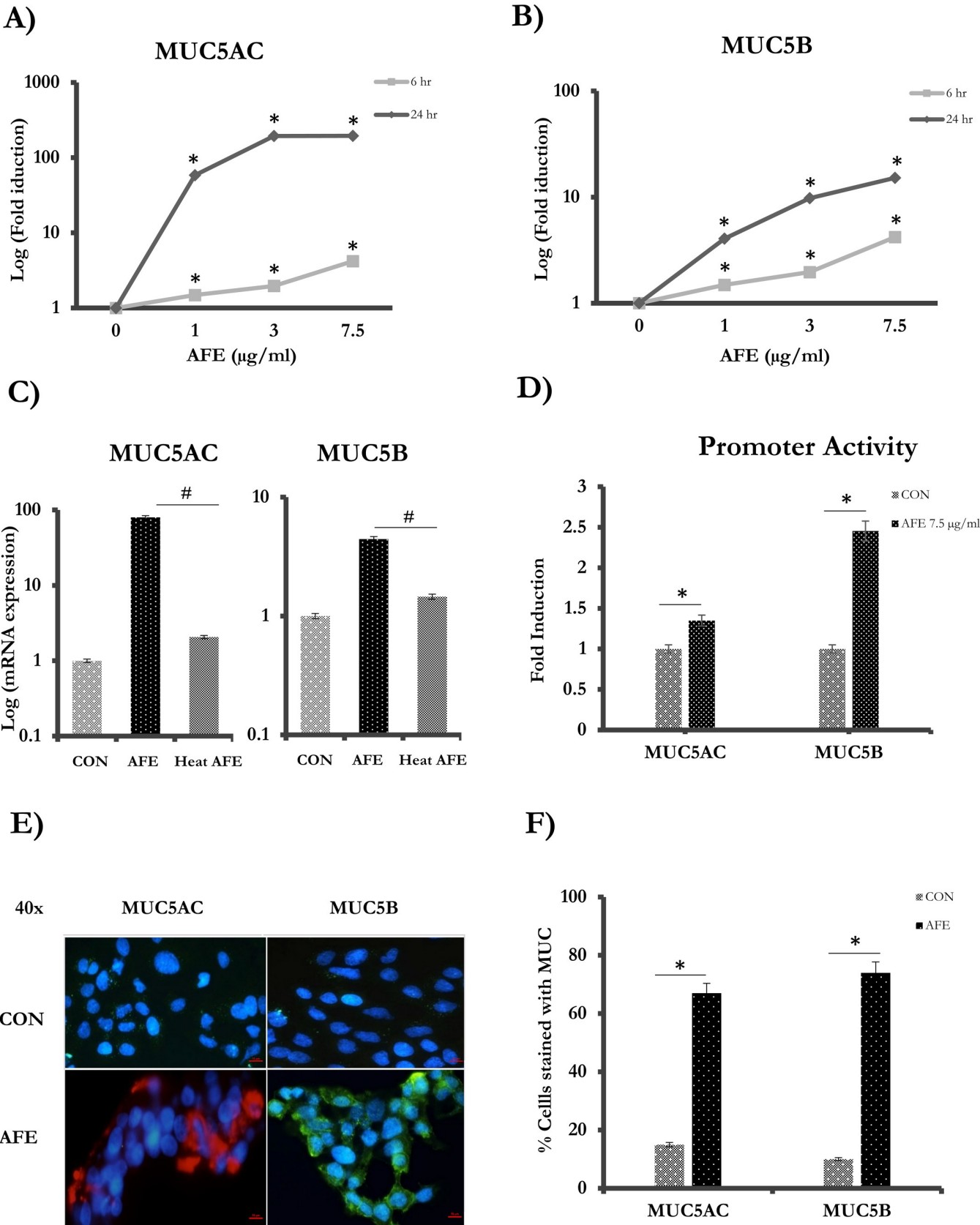

**Fig 1. Mucin induction by AFE treatment.** Cells were treated with different concentrations (1, 3 and 7.5 µg/ml) of AFE. RNA was isolated at 6 hrs (gray) and 24 hrs (black) later for Real-time PCR analysis for (A) MUC5AC and (B) MUC5B. (C) Cells were treated with 7.5 µg/ml of AFE or the same AFE but was heated at 72˚ C for 30 min. Samples were collected at 24 hrs later and analyzed for mucin expression. (D) Cells were transfected with MUC5AC or MUC5B promoter, and then treated with AFE for 24 hrs. Dual luciferase reporter assay was used to measure the activity. (E) Cells with or without AFE treatment were stained with anti-MUC5AC or anti-MUC5B antibody. Red: MUC5AC. Green: MUC5B. Blue: DAPI staining for nucleoli. Scale bar = 10 µm. (F) Quantification of Immunofluorescence-positive cells with or without AFE treatment. [#], [*]: $P < 0.05$.

folds, respectively (Fig 1D). This is in contrast to the mRNA measurement, in which the magnitude of MUC5AC mRNA increase was much greater than that of MUC5B. The precise nature of this discrepancy is unclear, but it is likely related to the incomplete cloning of a promoter region, where some of the positive elements may not be included. Nonetheless, AFE-induced increase of mucin mRNA was at least partly reflected by the increase of mucin promoter activity, suggesting that transcription activation by AFE was responsible for the increase of mucin mRNA to a certain extent.

To measure mucin protein production, cells were stained with specific anti-MUC5AC [29] or anti-MUC5B antibody [30]. Consistently, both MUC5AC and MUC5B were significantly elevated by AFE treatment (Fig 1E). Nearly 65% of cells expressed MUC5AC and 70% expressed MUC5B when stimulated with AFE. In contrast, less than 10% of the cells expressed these proteins at the baseline (Fig 1F). Therefore, AFE significantly increased the productions of major airway mucins-MUC5AC and MUC5B at both mRNA and protein levels.

## The basal EGFR activity is required for AFE-induced mucin expression

EGFR pathway is well established to mediate MUC5AC induction by a variety of treatments [31]. In the classical EGFR signaling pathway, the binding of ligands activates EGFR through its autophosphorylation, further activating its downstream signaling pathway to increase mucin production[32]. In order to test whether AFE-induced mucin expression was mediated by EGFR activation, the cells were pre-treated with each of the two selective EGFR tyrosine kinase inhibitors-BIBX1382 and AG1478. Indeed, AFE-induced MUC5AC mRNA expression was repressed by greater than 80% under the treatment of either of these inhibitors (Fig 2A). For MUC5B, the expression was repressed by 60% when the cells were treated with BIBX1382, or 43% when treated with AG1478 (Fig 2B). AG1478 had no cellular toxicity across the entire dose range as demonstrated by a cell viability assay (S1A Fig). To exclude the possibility of off-target effect of a chemical inhibitor, we blocked EGFR using an anti-EGFR neutralizing antibody, which led to a 67% decrease of AFE-induced MUC5AC expression (Fig 2C) and a 50% decrease of MUC5B expression (Fig 2D), respectively. Thus, based on these inhibition studies, EGFR pathway appeared to mediate AFE induced mucin expression. However, to our surprise, when we tried to confirm if EGFR was indeed activated by measuring the level of phosphorylated EGFR, we found that AFE did not increase EGFR phosphorylation at its major autophosphorylation site-Tyrosine 1173 (Fig 2E) [33, 34]. Because EGFR has several different tyrosine residues, we further tested total level of phosphor-tyrosine in EGFR by using immune-precipitation. Consistently, we could not detect any increase by AFE treatment (Fig 2F). Thus, even though EGFR activity was required for mucin induction, AFE treatment did not increase its activity. To exclude the possibility that these cells might be defective in further enhancement of EGFR signaling, we treated cells with EGF, the cognate ligand of EGFR. EGF treatment markedly increased EGFR phosphorylation (Fig 2G), and also significantly elevated expressions of MUC5AC and MUC5B (Fig 2H). Thus, these cells were able to respond to EGFR stimulus (e.g. EGF) by increasing EGFR phosphorylation leading to the increase of mucin gene expression. Therefore, we found that, although the basal activity of EGFR was required for mucin induction, AFE appeared not to significantly increase EGFR phosphorylation.

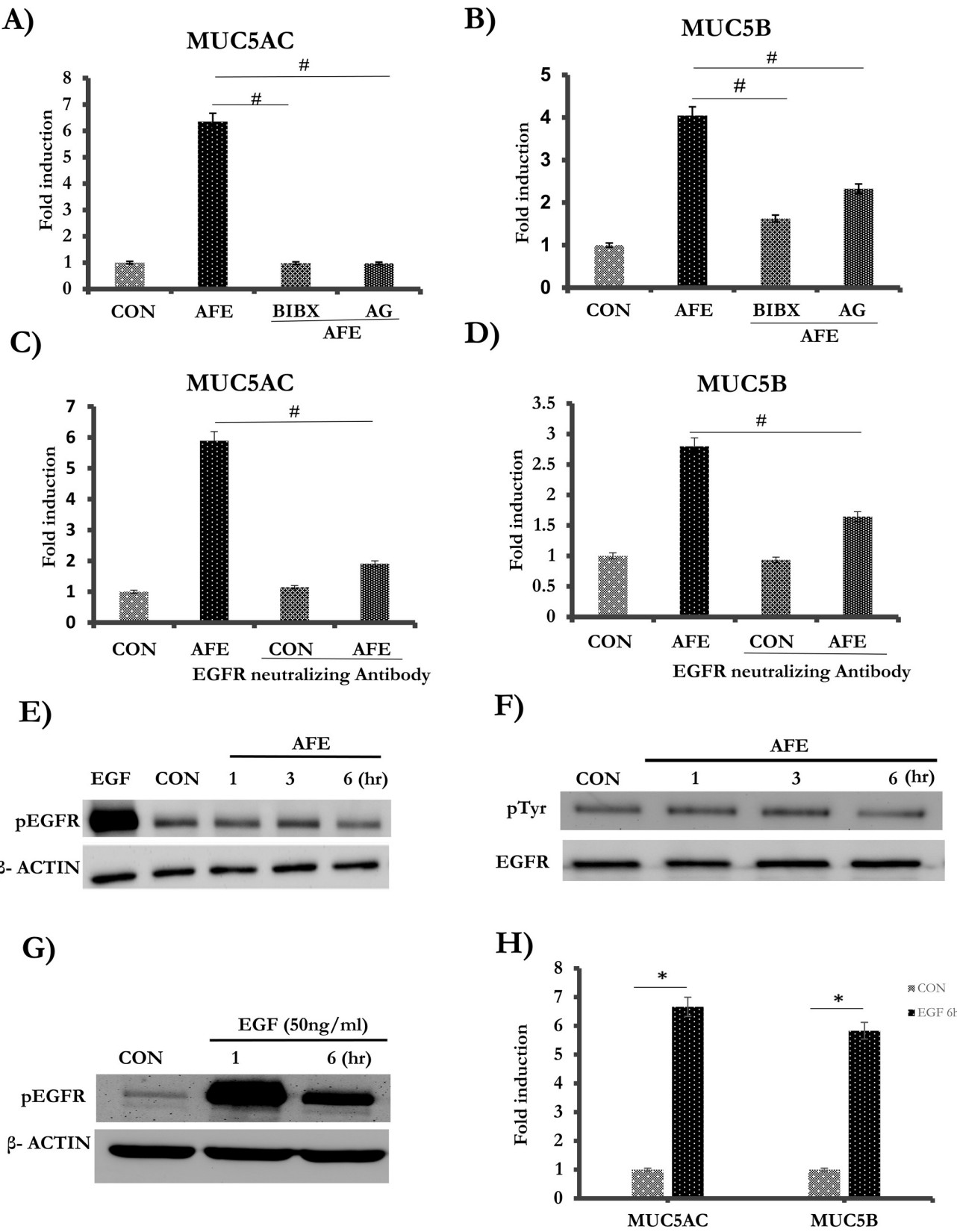

**Fig 2. The role of EGFR on AFE induced mucin expression.** The cells were pre-treated with EGFR inhibitors, BIBX1382 (5 μM), AG1478 (2 μM) for 1 hr, and then treated with 7.5 μg/ml of AFE for 6 hrs. (A) MUC5AC and (B) MUC5B were quantified by Real-Time PCR. (C-D) The cells were pre-treated with 1 μg/ml of neutralizing anti-EGFR antibody and then AFE stimulation for 6 hrs. (C) MUC5AC and (D) MUC5B were measured. (E) The cells were stimulated with EGFR (positive control) and 7.5 μg/ml of AFE for 1, 3, 6 hrs and then probed with pEGFR and β- ACTIN. (F) The cells were stimulated with 7.5 μg/ml of AFE for 1, 3, 6 hr, and then immuneprecipitated with EGFR antibody. pEGFR was blotted with phosphotysine antibody (pTyr). Total EGFR was also blotted for the input control. (G) The cells were stimulated with 50 ng/ml of EGF for 1 or 6 hrs, and then probed with pEGFR and β- ACTIN. (H) The cells were stimulated with 50 ng/ml of EGF for 6 hrs. and then harvested for Real-time PCR analysis for MUC5AC and MUC5B. [#, *]: $P < 0.05$.

## Fungal protease activity was required for mucin expressions

AFE is the mixture of variety of substances secreted by *A.fumigatus* [35, 36]. Furthermore, fungi have been known to secrete abundant proteases, that are harmful factors, into the environment [37]. In our study, the mucin-inducing activity in AFE appeared to be heat labile (Fig 1C), implying that enzymes such as proteases might be involved. Therefore, we tested if protease activity could indeed be detected in AFE, using a fluorescence-based protease assay. AFE was found to contain nearly 246.9 μg (trypsin equivalence) /g serine protease activity. To test if the protease activity was required for the increase of mucin gene expression, we treated cells with trypsin for 6 hrs. Indeed, both MUC5AC and 5B could be increased by trypsin treatment alone (Fig 3A). Furthermore, PMSF, a serine protease inhibitor, suppressed AFE-induced MUC5AC (Fig 3B) and 5B mRNA (Fig 3C). Serine proteases have been well established to activate PAR2 (Protease-Activated Receptor-2), and PAR2 mediates many physiological effects of exogenous and endogenous proteases such as trypsin [38]. Serine proteases from Alternaria, another fungal allergen, were reported to activate airway epithelial cells and induced pulmonary inflammation through PAR2 [39]. Thus, we tested the role of PAR2 in AFE-induced mucin expression. Interestingly, both low- (2 μg/ml) and high- (20 μg/ml) dose of anti-PAR2 neutralizing antibody failed in blocking mucin gene expression (Fig 3B and 3C), although the antibody did block the effect of PAR2 ligand (i.e. 2-Furoyl-LIGRLO-amide). Thus, AFE proteases stimulated airway mucin expression likely through a PAR2 independent pathway. Furthermore, we tested downstream signaling events by treating the cells with trypsin for 1 and 6 hrs. Similar to AFE (Fig 2F), trypsin did not enhance (actually decreased) EGFR phosphorylation (Fig 3D). Instead, trypsin markedly increased activated Ras and phosphorylated ERK1/2, suggesting that Ras/ERK activation might mediate protease-induced mucin expression. Trypsin had no cellular toxicity across the entire dose range as demonstrated by a cell viability assay (S1B Fig)

## AFE protease induced Ras/Raf1/ERK signaling pathway was responsible for enhanced mucin expressions

Because trypsin was able to activate Ras/ERK (Fig 3D), we examined if this pathway was also activated by AFE. AFE activated Ras/Raf/ERK as early as 1 hr after the treatment. The Ras activation peaked at 3 hrs and then returned to baseline at 6 hrs, which was different from a persistent activation under trypsin treatment (Fig 4A). In contrast, Raf1 and ERK were consistently activated through 6 hrs after the treatment (Fig 4B). To confirm the hierarchy of the signaling cascade, we utilized established inhibitors of Raf1 and ERK. Indeed, ERK activation was repressed by two Raf1 inhibitors: Raf1 Kinase Inhibitor I and sorafenib (Fig 4C). The controls were DMSO (solvent of Raf-1 inhibitor I) and $H_2O$ (the solvent of sorafenib). Both Raf1 and ERK (U0126) inhibitors almost completely repressed expressions of MUC5AC (Fig 4D–4F) and MUC5B (Fig 4G–4I). Therefore, AFE appeared to increase mucin gene expression through a Ras/Raf/ERK pathway. DMSO had no cellular toxicity across the entire dose range as demonstrated by a cell viability assay (S1C Fig).

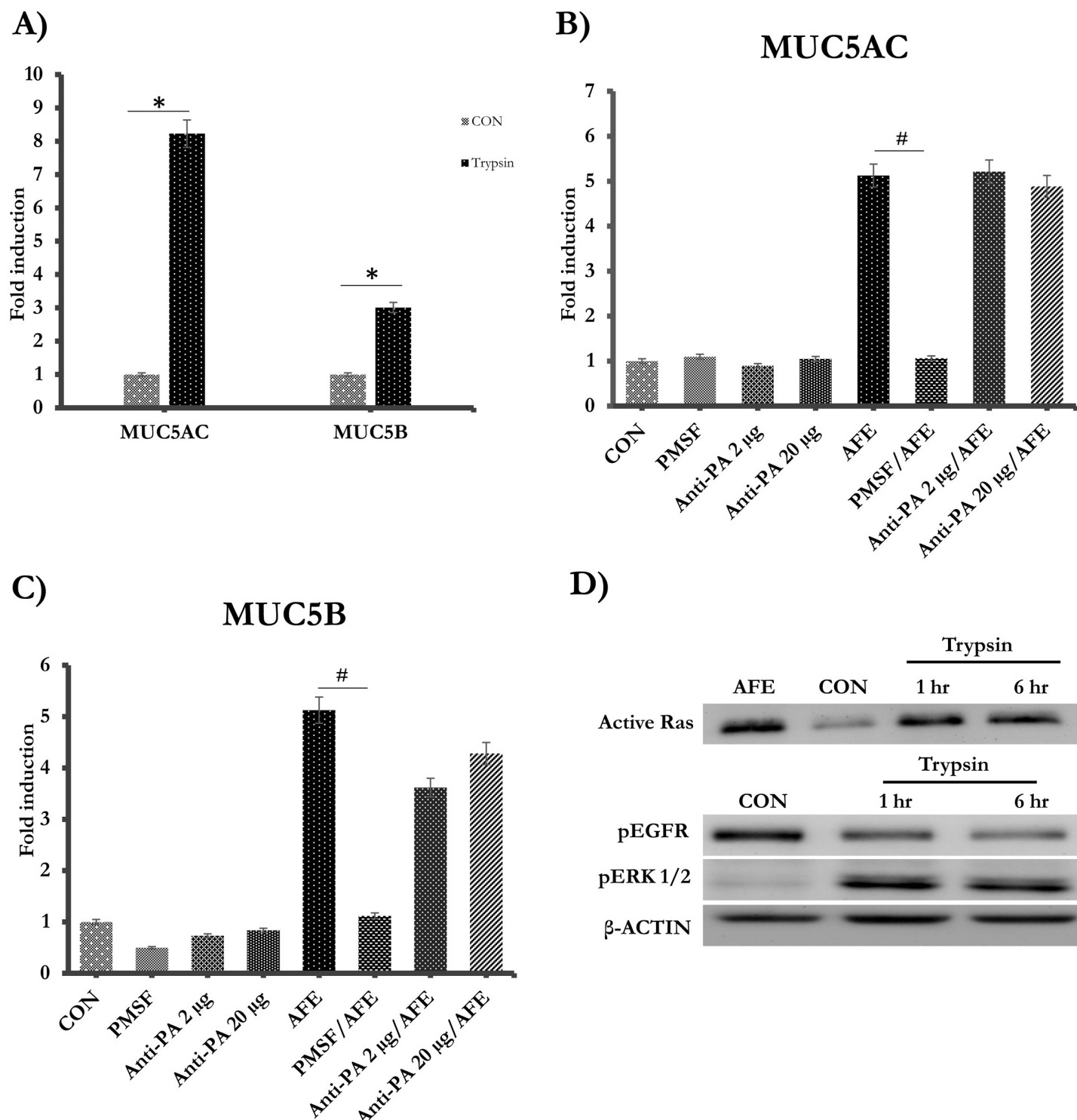

**Fig 3. Fungal protease activity was required for mucin expressions.** (A) The cells were treated with 100 nM trypsin for 6 hrs. MUC5AC and MUC5B were quantified by Real-Time PCR. (B-C) The cells were pre-incubated with two different doses (2 μg/ml and 20 μg/ml) of PAR2 antibody and then stimulated with AFE for 6 hrs. MUC5AC and MUC5B mRNA levels were measured. (D) The cells were treated 100 nM trypsin for 1 hr, 6 hrs and then probed with anti-Ras, pEGFR, pERK1/2 and β-ACTIN antibody. $^{\#,\ *}P < 0.05$.

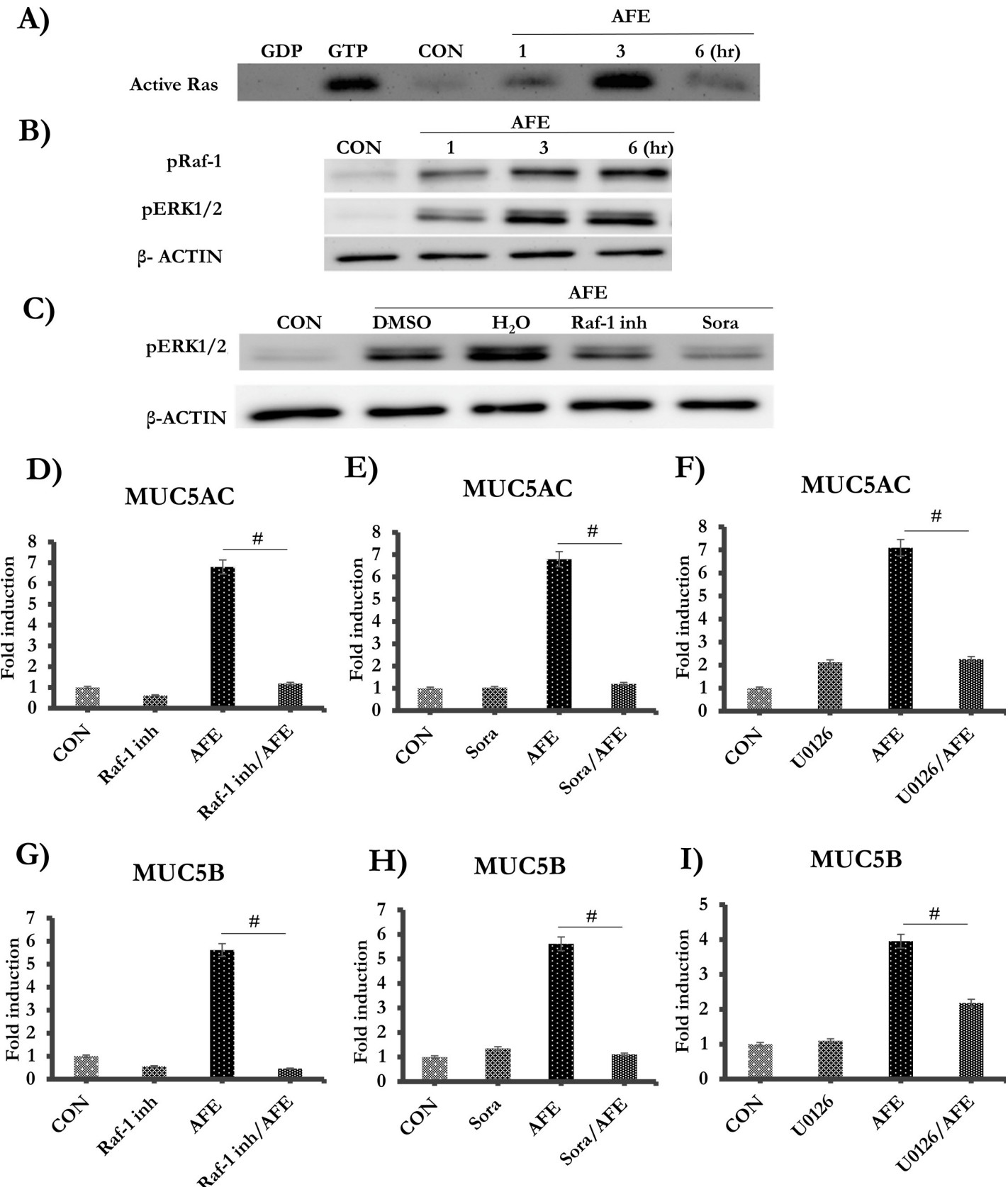

**Fig 4. AFE induced mucin production was mediated by a Ras/Raf1/ERK pathway.** The cells were stimulated with 7.5 μg/ml AFE at 1, 3, and 6 hrs. (A) Equal amount of the protein from each treatment was subject to active Ras pull-down assay to detect 21 kDa active Ras. A positive (GTP) and a negative (GDP) control were included. (B) Western blot analysis for pRaf1, pERK1/2 and β- ACTIN. (C) Cells were pre-treated with Raf-1 inhibitor I (10 μM) or Sorafenib (Sora, 40 μM) for 1 hr, then treated with AFE for 6 hrs. Western blot analysis was performed for pERK1/2 and β- ACTIN. The cells were pre-treated with Raf-1 inhibitor I, Sorafenib or U0126, followed by AFE treatment. Real-time PCR analysis was performed for MUC5AC (D-F) and MUC5B (G-I). #: $P < 0.05$.

Reactive oxygen species (ROS) is one important second messengers induced by AFE protease treatment [40], and ROS has also been linked to mucin induction [41, 42]. According to microscopic fluorescence imaging with ROS indicator (CM-H2DCFDA), AFE induced robust ROS production that lasted for at least 20 min and the faded (S2A and S2B Fig). Application with GSH-MEE [43], a cell permeable compound that can be readily converted to GSH-one of the most potent antioxidants, significantly blocked the dose-dependent AFE-driven ROS generation (S2C Fig). However, the repression of ROS generation doesn't affect the mucin expression (S2D Fig).

Ca$^{2+}$ flux was another important second messenger stimulated by AFE proteases [44]. According to the Ca$^{2+}$ imaging, AFE robustly induced Ca$^{2+}$ flux by up to 10-fold (Fig 5A and 5B). When Ca$^{2+}$ flux was blocked by BAPTA, a calcium chelator, AFE-induced mucin expression was significantly decreased (Fig 5C), suggesting that Ca$^{2+}$ was involved in mucin regulation by AFE. Interesting, Ca$^{2+}$ flux was shown to activate ERK signaling pathway [45, 46]. Indeed, we found that ERK activation by AFE was significantly repressed by BAPTA (Fig 5D). In contrast, the levels of activated Ras (Fig 5E) and Raf1 (Fig 5D) were enhanced. Thus, Ca$^{2+}$ flux likely had opposite impact on ERK and Ras/Raf1 with positive effect on ERK but a negative effect on Ras/Raf1. Nonetheless, the overall effect of Ca$^{2+}$ was to enhance ERK and mucin induction, which was perhaps due to the fact that ERK was downstream of Ras/Raf1.

## Discussion

Environmental mold exposure constitutes a significant threat to human health. The effects of mold exposure range from allergy and asthma to disseminated infections [47–49]. In the case of *Aspergillus*, although diseases caused by opportunistic infection such as aspergilloma and invasive pulmonary aspergillosis are devastating and sometimes lethal, they rarely occur in healthy individuals [1, 50, 51]. It is believed that a number of innate and adaptive defense systems including mucociliary escalator, phagocytes, and adaptive immune cells are in place to control the infection and to facilitate fungal clearance [52–55]. In contrast to these invasive diseases, mold induced hypersensitive diseases such as fungal asthma pose a much larger threat to the general public, as nearly 50% of otherwise healthy individuals develop fungal allergy during their lifetime [56]. Consistently, the prevalence of fungal sensitization is up to 48% in asthmatics [9]. The overall prevalence of asthma has significantly increased in United States and in other industrialized countries [57]. Asthma is a chronic airway disease characterized by chronic lower-airway inflammation, mucous cell metaplasia (MCM), and airway hyperresponsiveness (AHR) [58]. "Severe Asthma with Fungal Sensitization" or SAFS has been coined for the severe asthma with the sensitization to *Aspergillus*, *Alternaria*, *Cladosporium* and/or *Penicillum* [10]. Due to its thermo-tolerance to the normal human body temperature [5], *A. fumigatus* is one of the most important human pathogens to cause fungal asthma.

The fundamental pathogenesis of asthma lies in the airway obstruction in this disease. Both AHR and physical occlusion contribute to the overall airway obstruction [21, 59]. Airway mucus overproduction induced by T2IR has been shown to be responsible for both AHR and occlusion of the airway, particularly at the small bronchus [18]. In human airway, constitutively expressed MUC5B mainly contributes to the maintenance of homeostasis of the airway at baseline, and highly inducible MUC5AC is likely to illicit additional protective function [21,

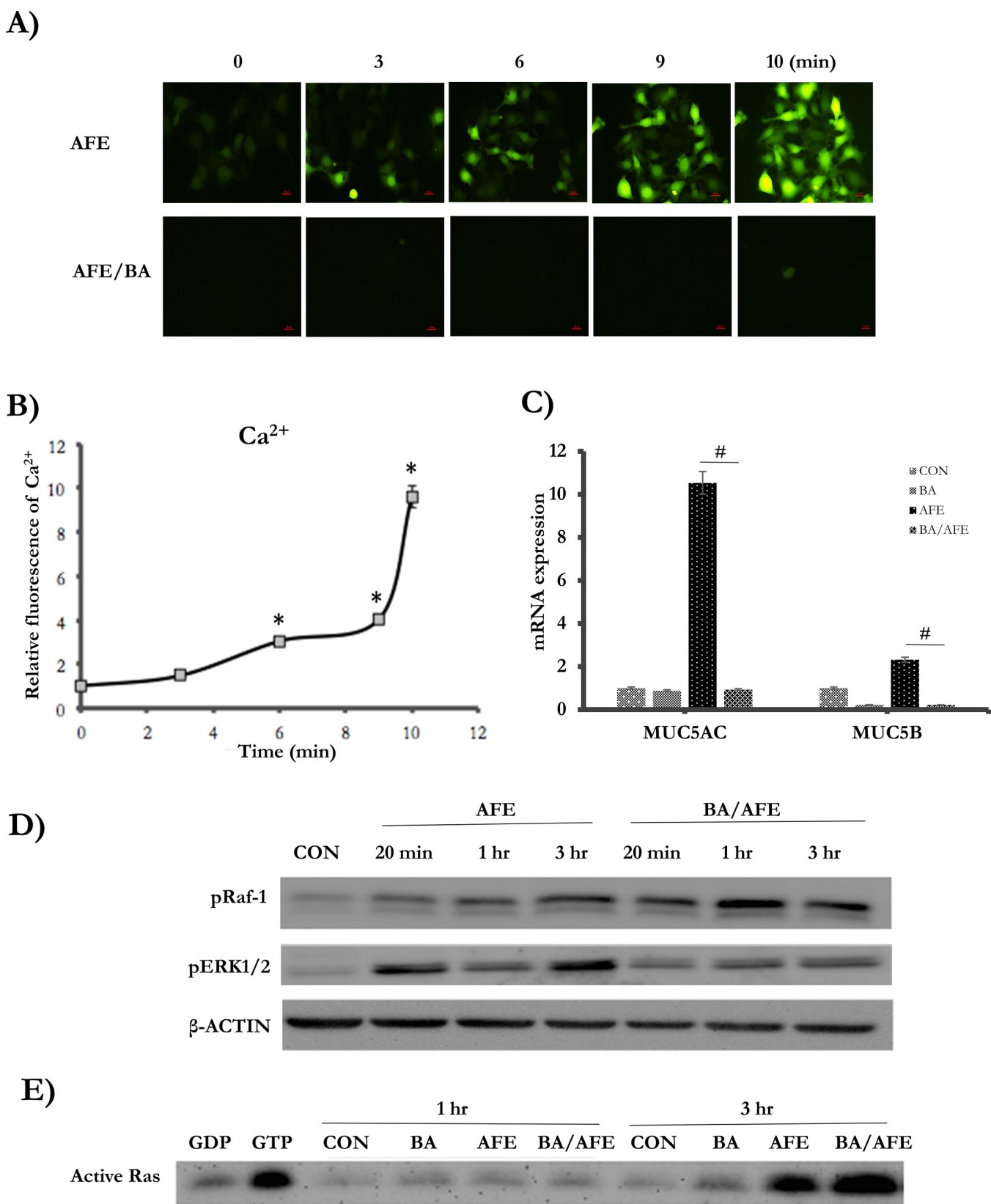

**Fig 5. AFE-induced Ca$^{2+}$ flux was indispensable for mucin induction by AFE.** (A) AFE induced Ca$^{2+}$ imaging was measured by staining with calcium indicator, fluo-4 AM. Ca$^{2+}$ flux could be blocked by pre-treated with 50 **μM** BAPTA-AM, a Ca$^{2+}$ chelator. Scale bar = 10 **μ**m. (B) Quantification of Ca$^{2+}$ flux induced by AFE treatment. $^*P < 0.05$. (C) The cells were pre-treated with 50 **μM** BAPTA-AM, followed by 7.5 **μg**/ml of AFE treatment for 6 hrs. MUC5AC mRNA levels were quantified by Real-Time PCR. $^{\#}P < 0.05$. (D) The cells were pre-treated (1 hr) with or without 50 **μM** BAPTA, and then stimulated with 7.5 **μg**/ml AFE for 20 min,1and 3 hrs. The samples were subject to western blot analysis for pRaf-1 and pERK1/2. B-ACTIN was used as a loading control. (E) The cells were pre-treated with or without 50 **μM** BAPTA-AM, and followed by 7.5 **μg**/ml AFE treatment for 1, 3 hrs. Equal amount of the protein from each treatment was subject to active Ras pull-down assay was performed. A positive (GTP) and a negative (GDP) control were included.

59]. Mucus overproduction is the hallmark of two major allergy illnesses (ABPA and fungal asthma) induced by *A. fumigatus*. In the animal model, chronic exposure to *A. fumigatus* was shown to induce MUC5AC expression in asthmatic rats [60]. In the present study, AFE, a crude preparation of *A. fumigatus* antigens, markedly elevated expressions of both MUC5AC and MUC5B in airway epithelial cells, suggesting a direct mucin-inducing effect of fungal substances in the absence of most of the other T2IR components such as mucin-inducing cytokines-IL1β, 6, 4, 9, 17 and 13[61]. The effects of enhanced mucin gene expression and protein production were likely protective in the acute phase as the mucociliary escalator is one of the most powerful innate defense mechanisms for environmental insults. Interestingly, serine protease secreted by *A.fumigatus* was shown to degrade airway mucus [62], likely utilizing these degraded polysaccharides and proteinaceous substrates as food sources [63, 64] or serving as a counter-measure. However, in the chronic phase, overproduced mucin proteins may obstruct the airway and induce AHR.

EGFR signaling has been a paradigm in the field of mucin regulation [31]. EGFR activation has been shown to mediate mucin, particularly MUC5AC, expression by a variety of stimuli such as bacteria or its components LPS, virus and its intermediates dsRNA, Elastase, H$_2$O$_2$ [32]. EGFR pathway was also found to mediate MUC5AC induction by AFE proteases [22] in NCI-H292 cells, commonly used *in vitro* epithelial cell model for mucin research [31]. However, no direct evidence of EGFR activation was demonstrated in that study, despite the efficacy of EGFR inhibition in repressing mucin gene expression. We initially planned to use this model to understand AFE-EGFR axis in mucin regulation. To our surprise, although basal level of EGFR activity was observed in these cells as demonstrated by EGFR phosphorylation, AFE treatment did not further enhance the level of phosphorylated EGFR at the common tyrosine residues (Tyrosine 1703). Thus, AFE appeared not to enhance EGFR activity as the other study implied but not demonstrated [22]. We also showed that these cells had intact EGFR signaling machinery, as EGFR signaling activation by its cognate ligand-EGF readily increased mucin gene expression. Therefore, our data support an alternate model, in which, EGFR activity was merely required but not sufficient for AFE-induced mucin gene expression.

Excitingly, we found AFE treatment dramatically increased active Ras and further enhanced its downstream Raf1-ERK cascade. This effect was likely mediated by serine proteases in AFE, as its mucin inducing activity was dependent on serine protease activity and the treatment of a model serine protease-trypsin alone activated Ras but not EGFR. Thus, Ras, but not EGFR, was the most likely target of AFE. PAR2, a G-protein coupled receptor (GPCR) was activated by proteases in a number of allergens in airway epithelial cells[39, 65–67], and was reported to activate Ras [67, 68]. However, we recently reported that allergen rich extracts from another import asthma-related fungus-Alternaria activated airway epithelial cells independent of PAR2 [25]. PAR2 independent pathway was also likely responsible for the specific effect of AFE on mucin gene expression, as the neutralizing antibody of PAR2 failed in preventing mucin induction by AFE. Other PARs (PAR1, 3 and 4) are expressed on epithelial cells [69]. Besides PAR2, PAR1 and 4 was shown to be activated by their ligands in airway epithelial cells [70]. However, their roles in the context of mold (fungal) proteases have not been well

understood and will need further investigation. Because $Ca^{2+}$ flux [44] and ROS [40] are two important second messengers downstream of *Aspergillus* proteases, we tested how they affected AFE-activated Ras/Raf1/ERK. Although both were found to be activated by AFE in our study, the inhibition of $Ca^{2+}$ flux, but not ROS, significantly repressed ERK activation and mucin induction. Paradoxically, the blockade of $Ca^{2+}$ flux was found to enhance AFE-induced Ras/Raf1. This discrepancy suggests that Ras/Raf1 was decoupled from ERK/mucin in the presence of $Ca^{2+}$ flux blocker. Interestingly, the decoupling between Ras/Raf1 and ERK was documented in the cancer field when the cells lost their adhesion[71]. As AFE proteases can cause cell detachment, perhaps the alteration of cell adhesion added another regulatory layer. Nonetheless, AFE serine proteases appeared to induce mucin gene expression *via* a Ras/Raf1/ERK cascade in airway epithelial cells, and $Ca^{2+}$ flux was an indispensable modulator of this process.

In summary, we have demonstrated that AFE potently induced expressions of MUC5AC and MUC5B in airway epithelial cells. Fungal serine proteases were responsible for these inductions but likely independent of PAR-2. EGFR activity was required but not sufficient for the mucin-inducing activity of AFE. Instead, fungal serine proteases activated a Ras/Raf1/ERK cascade further leading to mucin induction. Blockade of $Ca^{2+}$ flux, but not ROS, prevented mucin induction by AFE. Further extention of this study to *in vivo* models will advance our understanding of mucin/mucus regulation by *A. fumigatus* and further establish a Protease/Ras/Raf1/ERK axis as a novel therapeutic target for treating lung diseases caused by environmental mold exposure.

## Supporting information

**S1 Fig. NCI-H292 cells viability when treated with AG1478, Trypsin and DMSO.**
(PDF)

**S2 Fig. ROS production was induced by AFE.**
(PDF)

**S1 Raw Images.**
(PDF)

## Author Contributions

**Data curation:** Xianxian Wu, Boram Lee.

**Formal analysis:** Boram Lee.

**Funding acquisition:** Yin Chen.

**Project administration:** Yin Chen.

**Supervision:** Lingxiang Zhu.

**Writing – original draft:** Xianxian Wu, Yin Chen.

**Writing – review & editing:** Zhi Ding.

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
