## [Decision Letter · Decision Letter 0]

6 Feb 2020

PONE-D-20-00929

Exposure to mold proteases stimulates mucin production in airway epithelial cells through Ras/Raf1/ERK signal pathway

PLOS ONE

Dear Dr. Chen,,

Thank you for submitting your manuscript to PLOS ONE. After careful consideration, we feel that it has merit but does not fully meet PLOS ONE’s publication criteria as it currently stands. Therefore, we invite you to submit a revised version of the manuscript that addresses the points raised by the reviewer.

We would appreciate receiving your revised manuscript by Mar 22 2020 11:59PM. To enhance the reproducibility of your results, we recommend that if applicable you deposit your laboratory protocols in protocols.io, where a protocol can be assigned its own identifier (DOI) such that it can be cited independently in the future. For instructions see: http://journals.plos.org/plosone/s/submission-guidelines#loc-laboratory-protocols

We look forward to receiving your revised manuscript.

Kind regards,

Hong Wei Chu

Academic Editor

PLOS ONE

Journal Requirements:

2. At this time, we ask that you please provide the product numbers and lot numbers of the A. fumigatus extracts purchased from GREER (Lenoir, NC), the inhibitors/neutralizzing antibodies and primary antibodies used in this study.

"The study was supported partly by NIH grant AI39439.

Reviewers' comments:

Reviewer's Responses to Questions

**Comments to the Author**

1. Is the manuscript technically sound, and do the data support the conclusions?

Reviewer #1: Partly

2. Has the statistical analysis been performed appropriately and rigorously? 

Reviewer #1: Yes

3. Have the authors made all data underlying the findings in their manuscript fully available?

Reviewer #1: Yes

4. Is the manuscript presented in an intelligible fashion and written in standard English?

Reviewer #1: Yes

5. Review Comments to the Author

Reviewer #1: The cells response to AFE in MUC5AC transcripts are more than 10 times higher in the 24 hours time-point. Please clarify why 6 hours time-point was chosen.

Is the concentration of AFE physiologically relevant during fungal infections? Please clarify why these doses were chosen.

It was mentioned earlier that the 6 hrs time-point was selected. Please clarify why showing the 24 hrs time-point in figure 1? Please quantify MUC5AC and MUC5B protein levels for figure 1.

What are the diluents of EGFR inhibitors?

Do cells survive for 24 hrs after the EGFR inhibitor treatment? An LDH cytotoxicity assay will determine how toxic these inhibitors are to the cells.

At which time-point cells were stained in figure 5 E?

A Dot Blot can be performed for figure 2 to determine the MUC5AC and MUC5B protein levels reduction in the presence of EGFR inhibitors.

LDH cytotoxicity assay should be performed for figure 3 to determine the toxicity of incubating cells with trypsin for 6 hrs.

Graph labels are not clear figure 3 B-C.

What percentage of DMSO was used in figure 4? Is this amount of DMSO toxic to cells? LDH cytotoxicity assay will answer this question.

Please clarify why MUC5B data is not shown in figure 5?

Please highlight the significance of this work and how it contributes to the asthma research field in the discussion section.

6. PLOS authors have the option to publish the peer review history of their article (what does this mean?). If published, this will include your full peer review and any attached files.

Reviewer #1: Yes: Nastaran Mues

---

## [Author Response · Author response to Decision Letter 0]

20 Mar 2020

Dear reviewer:

Thank you for your advice, it was very helpful for me. Here are my answers. Reply=R

Response for first review: 

1. Please ensure that your manuscript meets PLOS ONE's style requirements. 

R: Yes, I have checked. My manuscript meets PLOS ONE's style requirements.

2. At this time, we ask that you please provide the product numbers and lot numbers of the A. fumigatus extracts purchased from GREER (Lenoir, NC), the inhibitors/neutralizzing antibodies and primary antibodies used in this study.

R: The product sheet is below.

3. Thank you for stating in your Funding Statement: "The study was supported partly by NIH grant AI39439.

Please also include the statement “There was no additional external funding received for this study.” in your updated Funding Statement.

R: Yes, I have corrected it in my cover letter and I have changed my Funding sourse. 

4.About the WB original data. In your cover letter, please note whether your blot/gel image data are in Supporting. Information or posted at a public data repository.

R: I have provided my original blot data via Submission System and corrected it in my cover. letter.

5. PLOS requires an ORCID iD for the corresponding author in Editorial Manager on papers submitted after December 6th, 2016. 

R: Here is my ORCID iD: https://orcid.org/0000-0002-2191-6873.

6. We note that you have included the phrase “data not shown” in your manuscript. Unfortunately, this does not meet our data sharing requirements. PLOS does not permit references to inaccessible data. 

R: Yes, I have already corrected it.

7、Reviewer #1: The cells response to AFE in MUC5AC transcripts are more than 10 times higher in the 24 hours time-point. Please clarify why 6 hours time-point was chosen.

R: For the mechanistic study, the early event is usually driven by a few immediate signaling pathways that are easy to dissect. When the exposure time goes on, secondary or tertiary events triggered by initial signaling may complicate the overall data interpretation. Therefore, we tend to choose the time points as early as the responses are demonstrated. 

8、Is the concentration of AFE physiologically relevant during fungal infections? Please clarify 

 why these doses were chosen.

R: We tested the dose response and found that this dose can induce high levels of mucin expression without outright cell death. In reality, the level of AFE exposure has been difficult to establish as its direct measurement is lacking. Nonetheless, AFE exposure model will lead to the discovery of mechanistic basis of Aspergillus induced illnesses. 

9、It was mentioned earlier that the 6 hrs time-point was selected. Please clarify why showing

the 24 hrs time-point in figure 1? Please quantify MUC5AC and MUC5B protein levels for. figure 1.

R: The 24hr time point is shown to demonstrate that this effect is time-dependent. The quantified protein levels are shown in Fig 1F.

10、What are the diluents of EGFR inhibitors? 

R: DMSO.

11、Do cells survive for 24 hrs after the EGFR inhibitor treatment? An LDH cytotoxicity assay

 will determine how toxic these inhibitors are to the cells.

R: Yes, cells still survive. Related data is in S1_File.

12、At which time-point cells were stained in figure 5 E?

R: I think you mean Fig 5A, cells were pre-treated with 50 uM BAPTA-AM for 30 min and. then used 5ul Fluo-4/AM in 1ml OPTI-MEM for another 30 min. After, wash 2x with OPTI-MEM (DMEM/HEPES/no phenol) and let it sit in fresh DMEM for another 30 minutes to allow the ester to cleave. At last, they were treated AFE for 3, 6,9, 10 min.

14、LDH cytotoxicity assay should be performed for figure 3 to determine the toxicity of 

incubating cells with trypsin for 6 hrs.

R: Related data is in S1_File.

15、Graph labels are not clear figure 3 B-C.

R: Thank you, I have corrected this.

16、What percentage of DMSO was used in figure 4? Is this amount of DMSO toxic to cells?

 LDH cytotoxicity assay will answer this question.

R: The maximum percentage is 0.3% for U0126 inhibitor in this experiment. Related data is in S1_File.

17、Please clarify why MUC5B data is not shown in figure 5?

R: Because 5B induction by AFE treatment is a little bit low, we didn’t show it at first, but I. have added this data in Fig 5C.

18、Please highlight the significance of this work and how it contributes to the asthma research

 field in the discussion section.

 R: Please see the highlight part in revised manusription.

Response to second review: 

1) Please provide additional information about each of the cell lines used in this work, including any quality control testing procedures (authentication, characterisation, and mycoplasma testing). For more information, please see http://journals.plos.org/plosone/s/submission-guidelines#loc-cell-lines.

R: I have add relevant information in Cell line section.

2) To comply with PLOS ONE submission guidelines, in your Methods section, please provide additional. information regarding your statistical analyses. For more information on PLOS ONE's expectations for statistical reporting, please see https://journals.plos.org/plosone/s/submission-guidelines.#loc-statistical-reporting.

R: I have corrected my statistical analyses in your Methods section.

3) At this time, we ask that you please add scale bars to the microscopy images presented in Figures 1. and 5 and refer to the scale bar in the corresponding Figure legend.

R: Yes, I have add the scale bar analyses in Figures 1. and 5.

 Product name Product number Lot number

 Aspergillus fumigatus XPM3C3A25 231795

 AG1478 658548-1MG D00108234 

 U0126 662005-1MG D00122919

 BIBX 1382 324832-5MG B77319

 neutralizing anti-EGFR Ab 05-101 0120201A1

 Raf-1 inhibitor 553003

 Sorafenib S-8502 BSF-106

 PMSF P7626-5G BCCB9214

 Glutathione reduced ethyl ester G1404-25MG BCCB5246

 PORCINE Trypsin 85450C-25MG 12N438AR1C

 EGF 01-407 DAM1687496

 BAPTA/AM 196419-25MG 3278031

 pERK1/2 (T202/Y204) Ab 9101S 15

 pRaf-1 (Ser289/296/301) Ab 9431S 2

 anti-PAR2 neutralizing (SAM11) Ab SC-13504 C0713

 anti-MUC5B (H-300) Ab SC -20119 F0611

 anti-pEGFR (Y1173) Ab SC -12351 E2813

 anti-EGFR (1005) Ab SC -03 H1007

 anti-pTyr (py99) Ab SC -7020 L1106

 β- ACTIN (C4) Ab SC -47778 H0515

 anti-MUC5AC Ab MS-145-PI 145P1405I

 EnzChek® Protease Assay Kit E6638 1445257

 RAS GTPASE PULLDOWN Kit PI16117 PA199259

 Promega* Dual-Glo* Luciferase Assay PR-E2940 0000303692 

 Cell Proliferation Assay Kit G3580 0000328878

 Fluo-4/AM F14201 2165131

---

## [Editor Report · Decision Letter 1]

6 Apr 2020

Exposure to mold proteases stimulates mucin production in airway epithelial cells through Ras/Raf1/ERK signal pathway

PONE-D-20-00929R1

Dear Dr. Chen,

We are pleased to inform you that your manuscript has been judged scientifically suitable for publication and will be formally accepted for publication once it complies with all outstanding technical requirements.

With kind regards,

Hong Wei Chu

Academic Editor

PLOS ONE
---

## [Editor Report · Acceptance letter]

10 Apr 2020

PONE-D-20-00929R1 

Exposure to mold proteases stimulates mucin production in airway epithelial cells through Ras/Raf1/ERK signal pathway 

Dear Dr. Chen:

I am pleased to inform you that your manuscript has been deemed suitable for publication in PLOS ONE. Congratulations! Your manuscript is now with our production department. 

With kind regards,

on behalf of

Dr. Hong Wei Chu 

Academic Editor

PLOS ONE